# The Past, Present, and Future of Cervical Cancer Vaccines

**DOI:** 10.3390/vaccines13020201

**Published:** 2025-02-17

**Authors:** Alexander C. Lien, Grace S. Johnson, Tianyun Guan, Caitlin P. Burns, Jacob M. Parker, Lijun Dong, Mark R. Wakefield, Yujiang Fang

**Affiliations:** 1Department of Microbiology, Immunology & Pathology, Des Moines University College of Osteopathic Medicine, West Des Moines, IA 50266, USA; alexander.c.lien@gmail.com (A.C.L.); caitlin.p.burns@dmu.edu (C.P.B.); jacob.parker@dmu.edu (J.M.P.); 2Department of Liberal Arts, Arizona State University, Tempe, AZ 85281, USA; gmjohn18@asu.edu; 3Department of Obstetrics and Gynecology, The Nanhua Hospital, Nanhua University, Hengyang 410004, China; dr.tianyunguanobgyn@outlook.com (T.G.); donglijunys@163.com (L.D.); 4Department of Surgery, University of Missouri School of Medicine, Columbia, MO 65212, USA; wakefieldmr@health.missouri.edu; 5Ellis Fischel Cancer Center, University of Missouri School of Medicine, Columbia, MO 65212, USA

**Keywords:** vaccine, cancer, HPV

## Abstract

Since the introduction of prophylactic HPV vaccines, both HPV infection rates and cervical cancer rates have subsequently dropped. Yet, cervical cancer remains the fourth most common cancer diagnosis in women globally. As HPV and its role in the development of cervical cancer become better understood, vaccines have emerged as a front runner for improved therapeutic cervical cancer treatment. Recent studies have shown that protein and DNA vaccines may be effectively delivered via the use of several different vectors, while combination therapy with immune checkpoint inhibitors provides even more effective treatment. Further investigation and additional clinical studies into specific vaccine strategies are necessary to determine how effective vaccines are as therapeutic treatment for cervical cancer. This review intends to summarize some of the most promising research on cervical cancer vaccines. Such a study may be helpful for gynecologists to prevent and manage patients with HPV infection.

## 1. Introduction

Human papillomavirus (HPV) infection is one of the most transmitted sexual diseases, with an estimated 13 million Americans becoming infected annually [1]. While most infections are cleared without treatment within two years [1], persistent HPV infections have been shown to cause a plethora of ailments, including cancers of the cervix. Approximately 90% of cervical cancer incidences are caused by HPV infections [2]. With an annual rate of around 660,000 new cases and 350,000 deaths globally, cervical cancer is a pertinent public health concern world-wide [3]. Fortunately, HPV has become largely preventable, and consequently, so has cervical cancer in many cases.

Since the introduction of prophylactic vaccines to the US in 2006, the prevalence of HPV infections, and subsequent cervical cancer diagnoses have been steadily falling [4,5]. Yet, cervical cancer remains the fourth most common cancer in women globally [6]. Though current HPV vaccines are highly effective, reducing the incidence of specific HPV strains by up to 90%, they are not perfect and do not confer complete immunity [7].

Furthermore, the highest rates of cervical cancer incidence are found in low- and middle-income countries [3]. The lack of access to HPV vaccinations and cervical cancer screening is the main culprit. This illustrates the socioeconomic determinants of HPV infections and, thus, cervical cancer rates [8].

Until HPV infections are eradicated globally, cervical cancer will continue to pose a severe public health issue and require improved cervical cancer treatment.

## 2. HPV

To date, more than 200 strains of HPV have been identified, the majority of which are categorized as low risk, meaning they are less likely to lead to cervical cancer [9]. Of the high-risk strains, HPV-16 and HPV-18 are responsible for 70% of cervical cancers [10]. The virus consists of two strands of circular papillomavirus, which can be further divided into three major sections: an early region (E), a late region (L), and a noncoding long control region (LCR) [9]. The early region (E1–7) plays a significant role in virus replication and carcinogenesis. The late region is composed of L1, a major capsid protein, and L2, a minor structural protein. The L1 protein has been shown to self-assemble into virus-like particles and is often the target of many prophylactic HPV vaccines [11]. Following HPV viral genome replication, the L1 and L2 genes are deleted [12]. Of the early-region genes, E6 and E7 play the largest roles in carcinogenesis [12]. E6 inhibits the tumor suppressor p53, while E7 inhibits the tumor suppressor Rb [13] Figure 1, allowing infected cells to avoid apoptosis and progress to a cancerous state. 

Additionally, HPV-infected cells are able to create an immunologically privileged site via the upregulation of immune checkpoints such as PD-L1 [14]. These immune checkpoints make it difficult for T-cells and other cancer-fighting cells to gain access to the infected site by binding to the PD1 portion of the T-cells, allowing for the unchecked spread of infection. The level of PD-L1 expression has been shown to vary among cervical squamous cell carcinoma and has prognostic significance [15]. Marginal PD-L1 expression in cervical carcinomas is likely induced by the signaling of interferon gamma, in comparison to more diffuse PD-L1 expression [16,17]. Interferon gamma is secreted from CD8 T-cells, whose cytotoxicity is crucial for the removal of both intracellular infections and malignant cells [18].

Once a tumor has been established, the tumor cells are also able to recruit immunosuppressive cells such as regulatory T-cells (Tregs) and myeloid-derived suppressor cells (MDSCs) [19]. These cells further alter the tumor microenvironment, allowing for continued tumor immune system evasion and promoting metastasis.

## 3. HPV Vaccines

Currently, six prophylactic HPV vaccines are approved for use [20] Table 1. GARDASIL^®^ (Hyderabad, India) and Cervarix^®^ (Wavre, Belgium) are the oldest, and perhaps the most well known. GARDASIL^®^ is a quadrivalent vaccine protecting against HPV-6, HPV-11, HPV-16, and HPV-18, while Cervarix^®^ is a bivalent vaccine protecting against HPV-16 and HPV-18. Both vaccines are highly effective in protecting against HPV infections [21,22,23,24,25]. These vaccines use the L1 portion of the HPV genome to induce the formation of HPV-specific antibodies, preventing infection after exposure. As previously stated, these vaccines are highly effective, with a maximal reduction of 90% of specific HPV strains [7]. However, they only protect against specific HPV strains and do not confer complete immunity. Additionally, they are only effective when administered prior to infection and are typically recommended for patients around the age of 11–12 years old in the clinic [4]. Furthermore, there is evidence suggesting that HPV vaccines have greater efficacy when administered to young patients (12–13 years old), and efficacy decreases significantly after the age of 18 [26]. Even though the introduction of prophylactic HPV vaccines has greatly reduced the incidence of both HPV infections and cervical cancer rates, it has not eliminated the problem. To address this gap, another possible approach, which requires further research, is using circulating miRNAs in diagnosis and therapy; this strategy has been investigated in endometrial cancer [27] and might prove valuable in the context of cervical cancer as well. Until HPV is completely eradicated, continued research into cervical cancer treatment is necessary.

## 4. Current Cervical Cancer Treatments

Current strategies for cervical cancer treatment are dependent on the stage of the cancer. These strategies include invasive surgeries, chemotherapy, radiation, or a combination of these [28]. Surgery is commonly used for early-stage cervical cancers that have not yet spread significantly. As with any surgery, there is potential for adverse outcomes, such as loss of patient fertility, among other common surgical risks [29]. For later-stage cervical cancers, chemotherapy, or chemotherapy alongside radiation, is considered the current standard of care [30]. Chemoradiotherapy is used when the cancer has not yet spread to other parts of the body, while chemotherapy alone is typically the treatment of choice for metastatic cervical cancer [31]. Cisplatin is the most commonly used chemotherapy drug to treat cervical cancer [32]. Chemotherapy is notorious for its side effects and adverse reactions. These include hair loss, fatigue, nausea and vomiting, and increased risk of infections, among many others [31].

Though treatment has come a long way for cervical cancer, it is still far from adequate. Current treatments come with a laundry list of potential adverse outcomes and side effects, while metastatic cervical cancer patients have a five-year relative survival rate of only 19% [33]. Such outcomes are unacceptable and highlight a pertinent need for continued improvement of treatment strategies.

On a more positive note, research is currently in progress for more novel treatment options for cervical cancer. In 2021, the Food and Drug Administration approved the use of pembrolizumab in combination with chemotherapy for the treatment of cervical cancer. Pembrolizumab is a humanized monoclonal anti-PD1 antibody that binds to T-cells, allowing the cells to bypass the PD-L1 immune checkpoint and invade neoplastic tissue [34]. Increased T-cell invasion allows for a more robust immune response to carcinomas, improving treatment outcomes for cancer patients.

## 5. Rationale for Vaccines as a Therapeutic Treatment

As previously stated, cervical cancer treatment is far from adequate and urgently requires improvement to minimize patient harm and improve outcomes. HPV can suppress tumor inhibitors and generate an immunologically privileged site. Immunotherapy and vaccine treatments offer the potential to develop treatments that can target these features of HPV.

The success of pembrolizumab, along with several new and ongoing studies, offer hope for the development of novel treatments to combat the public health crisis that is cervical cancer.

## 6. Accum™ and Vaccines

A hurdle to many cancer treatments is the ability of cancer cells to sequester antibody-drug conjugates within their endosomes and subsequently pump the contents of these endosomes into the extracellular space [35]. This process reduces the concentration of the therapeutic substance within cancer cells, reducing its effectiveness. The newly developed Accum™ (Vancouver, BC, Canada) allows for endosomal escape and nuclear targeting by using cholic acid and nuclear localization signaling [36]. The result is increased intracellular accumulation of the desired substance.

Bikorimana et al. investigated the use of an Accum-E7 (aE7)-based protein vaccine as a mechanism of delivering proteins to antigen-presenting cells and eliciting an immune response against cancer cells [37].

The study first explored the use of aE7 prophylactically. Mice were treated with either an aE7 vaccine, an E7 vaccine (no Accum™ technology), or saline as a control group. All treatments were administered alongside a Montanide™ ISA720 (Fontenay-aux-Roses, France) adjuvant. Two weeks after vaccination regiment completion, all mice were challenged with a cervical cancer cell line, and tumor growth was measured over time. The aE7 group was further challenged with additional cervical cancer cell lines at 30 and 60 days following vaccination to strengthen successful results. A parallel study was conducted testing five different aE7 dosage amounts.

The data from the prophylactic study suggest that aE7 vaccination was far superior to E7 vaccination alone. Both the E7-treated mice, and the control group showed consistent tumor growth over time, whereas the aE7 group never developed tumors. Additionally, the additional challenge with cervical cancer cell lines to the aE7 group was unable to establish tumor growth, suggesting lasting immunity. The dosage study showed that lower doses of aE7 were unable to completely restrict tumor growth, but adequate dosage kept the mice tumor-free for the entire three-month observation period. These results suggest that aE7, administered alongside an adjuvant, has the potential to mount an effective immune response capable of preventing cervical cancer establishment.

Following the prophylactic study, a therapeutic study was completed to assess the effectiveness of treatment against established cancer. All mice were challenged with a cervical cancer cell line. Five days after the challenge, or whenever a palpable tumor was recognized, the mice were again separated into three groups and treated with either aE7, E7, or saline, all mixed with adjuvant. A dosing experiment was again completed in parallel. The mice were then followed, and tumor growth was measured over time. To investigate therapeutic vaccination in combination with immune checkpoint inhibitors, the same experiment was conducted while also administering the mice anti-PD-1, anti-CTLA4, and anti-CD47 injections.

Considering that vaccine therapy will most likely be used in combination with immune checkpoint inhibitors, the cervical cell cancer line that the mice were challenged with was assessed for the expression of PD-L1, CTLA4, and CD47, and it was determined that CD47 had the strongest expression. When analyzing the data following the therapeutic study, it was shown that aE7 limited tumor cell growth in all cases. Cell growth was limited most when aE7 was combined with anti-CD47, which correlates with the strong expression of CD47 in the cervical cancer cell line. The dose dependence study showed that increased E7 dosage used in combination therapy was more effective in limited tumor cell growth. When considering mice with previously established tumor growth, the only group in the study that showed 100% survival in the following 40 days were the mice treated with aE7 and immune checkpoint inhibitors.

A Good Laboratory Practices (GLP) study was conducted by a third party, investigating the potential for adverse side effects. The results showed that regardless of dosage and sex, mice maintained their weight and did not show any gross pathology upon observation. It was also determined that there is likely a plateau in the effectiveness of dosage; however, as previously stated, increased dosage was not seen to be harmful.

Bikorimana et al.’s work has shown the potential for the effective use of Accum™ technology as a cervical cancer treatment, both prophylactically and therapeutically [37]. This is likely due to the ability of Accum™ to increase the accumulation of vaccine proteins in antigen-presenting cells, allowing for stronger and longer-lasting protection. Additional work needs to be performed to investigate the effectiveness of aE7 against other lines of cervical cancer and in combination with other immune checkpoint inhibitors, but the current results remain very promising.

## 7. DNA Vaccines and Adenovirus Vectors

DNA vaccines use an altered amino acid sequence of part of the viral genome to introduce a safe antigen, which the immune system can identify and form a protective response to [38]. Han et al. designed an adenoviral vector encoding a mutant HPV-16 E7 (Ad-E7) [39]. The human immune system generates a high number of neutralizing antibodies against certain adenovirus strains, in this case, adenovirus serotype 5 [40]. This makes an adenovirus recombinant an ideal vector for delivering a therapeutic vaccine against cervical cancer.

The mutant HPV-16 E7 used in the study had a deletion of amino acids 21–24, which resulted in the loss of Rb binding, limiting HPV’s capability of inhibiting the Rb tumor suppressor. To test the effectiveness of the Ad-E7 vaccine, mice were injected with a cervical cancer cell line and selected for tumor growth. The mice were then divided into groups and administered different combinations of treatments, including a control group; a group that just received Ad-E7; a group that just received an immune checkpoint inhibitor, either PD-1 or PD-L1 antibodies; and groups that received a combination of the Ad-E7 vaccine and either the PD-1 or PD-L1 immune checkpoint inhibitor. The mice were observed over time, and tumor growth was recorded every three days.

The results showed that treatment with just immune checkpoint inhibitors had no impact on curbing tumor growth when compared with the control groups. However, mice that received the Ad-E7 vaccine experienced a suppression in tumor growth, with tumors beginning to regress at day 15 and gradually subsiding as treatment continued. Additionally, mice that received the Ad-E7 vaccine in combination with PD-1 or PD-L1 showed a more significant decrease in tumor size, with tumors being barely detected after day 12.

To continue investigating the effectiveness of all treatment groups, the mice were euthanized, and spleen lymphocytes were observed. The results showed that mice treated with the Ad-E7 vaccine had an increase in both CD8 T-cells and interferon gamma levels in comparison to the control groups. Furthermore, mice treated with Ad-E7 in combination with an immune checkpoint inhibitor had an even greater increase in both CD8 T-cells and interferon gamma levels. Both Treg and MDSC levels were simultaneously measured at this time. Although the Ad-E7 vaccine alone did not appear to affect Treg and MDSC levels in comparison with the controls, Ad-E7 in combination with immune checkpoint inhibitors was able to lower Treg and MDSC levels.

Since tumors were no longer observable in mice treated with Ad-E7 at the time of euthanasia, the experiment was repeated, but with all mice being euthanized on day 14. Tumors were then observed to investigate the level of infiltration of CD8 T-cells and interferon gamma levels within the tumors. Again, mice treated with Ad-E7 in combination with immune checkpoint inhibitors showed marked increases in both CD8 T-cells and interferon gamma levels within the tumor in comparison with the control groups. Tumor Treg and MDSC levels were also measured and shown to be markedly lower in the Ad-E7 in combination with immune inhibitor groups, as expected. All these results suggest that Ad-E7 used in combination therapy confers an ability to the immune system to limit tumor growth by lowering Treg and MDSC levels while minimizing tumor size via invasion of CD8 T-cells and their corresponding release of interferon gamma. This correlates with the observed tumor sizes of the first experiment.

Tumors were further analyzed using H&E staining to assess structural differences as a result of treatment. A thicker tumor capsule differentiates the tumor from the surrounding tissue and is correlated with a lower risk of metastasis [41]. Lower capillary density is also correlated with a lower risk of tumor growth and metastasis [42]. Upon observation of the H&E stains, it was noted that mice in groups treated with Ad-E7 alone showed an increase in capsule thickness and a lower capillary density when compared to the control groups. Mice treated with Ad-E7 in combination with immune checkpoint inhibitors showed an even greater capsule thickness and an even lower capillary density than all other groups.

The results of the experiments conducted by Han et al. show great promise for the use of DNA-based vaccines as a therapeutic treatment for cervical cancer [39]. By using an adenovirus vector, it was shown that their vaccine can hinder tumor growth and lead to tumor recession. These results were further exaggerated when used in combination therapy, offering optimistic potential for the development of new treatments, tailored specifically to a patients’ tumor specifications [39]. Additional work needs to be performed to investigate whether similar DNA vaccines can be effective at treating strains of HPV other than HPV-16.

## 8. Cervical Precancerous Vaccines Used in Combination Therapy

The DNA vaccine GX-188E (tirvalimogene teraplasmid) is a therapeutic vaccine for precancerous cervical lesions that encodes the E6 and E7 proteins for both HPV-16 and HPV-18 [43]. Studies have shown that patients with cervical precancerous lesions may benefit from immunization with the GX-188E vaccine [44,45]. This is likely due to the increased HPV-specific interferon gamma responses associated with GX-188E vaccination. Multiple studies exploring the best use of GX-188E as an HPV treatment are ongoing.

In their study, Youn et al. investigated whether patients with recurrent or advanced cervical cancer would benefit from treatment with the GX-188E vaccine in combination with the immune checkpoint inhibitor pembrolizumab [46]. Their current results are based on an interim activity assessment. Patients were recruited based on a diagnosis of recurrent or advanced cervical cancer that was HPV-16- or HPV-18-positive. The participating patients received an intramuscular injection of GX-188E at weeks 1, 2, 4, 7, 13, and 19, with an optional dose to be given at week 46 at the investigator’s discretion. Intravenous pembrolizumab was administered every three weeks throughout the study for up to two years. The results were recorded at week 24, to judge the current efficacy of the study.

The patients were radiologically assessed to track their treatment response, using CT with contrast or contrast-enhanced MRI when CTs were contraindicated. The radiological assessments were conducted at nine-week intervals. Peripheral blood samples were also taken at baseline and weeks 1, 4, 7, 10, 16, and 22 to measure patient interferon gamma levels in response to treatment.

An interim assessment was completed to determine how many patients had experienced a complete or partial response to treatment by 24 weeks. The responses were evaluated based on the level of tumor regression. The results showed that 42% of patients experienced at least a partial response to treatment, with 15% experiencing a complete response. It is important to note that this was an interim assessment and the results were not centrally reviewed.

Youn et al.’s work has shown the potential for using cervical precancerous vaccines in combination therapy for treating advanced cervical cancer [47]. Such results open a new avenue for continued investigation into new vaccine therapies that could be used as therapeutic treatments for cervical cancer.

## 9. Lacticaseibacillus Oral Vaccine

Most vaccines, especially those being investigated as a therapeutic approach to treating HPV, are administered intramuscularly. Kawana et al. took a novel approach and investigated the potential of a therapeutic HPV-16 E7-expressing *lacticaseibacillus*-based oral vaccine [47].

The investigators designed a study centered around IGMKK16E7, a lacticaseibacillus paracasei-expressing cell surface, full-length HPV-16 E7 vaccine that is designed to be administered orally. In the double-blind, placebo-controlled, randomized trial that followed, patients were selected for HPV-16-positive high-grade cervical intraepithelial neoplasia 2 and 3. These patients were then assigned to one of four treatment groups: placebo, or low, intermediate, or high doses of IGMKK16E7. All four groups were given the oral IGMKK16E7 treatment at weeks 1, 2, 4, and 8. 

The high-dose group demonstrated a 19% greater rate of histopathological regression to normal compared to the placebo group. Additionally, for patients positive for HPV-16 only, there was a 28% increase in regression to normal in comparison to the placebo group. Regression-to-normal rates were observed in a dose-dependent manner. Patients with higher levels of regression were noted to have higher levels of interferon gamma-producing cells circulating in their peripheral blood. The rate of adverse events related to treatment was the same for the placebo group and high-dose group. 

Kawana et al.’s work found success, though limited in comparison to previously observed studies [47]. These results show that an oral vaccine may be possible by treating cervical cancer precursors such as E7. IGMKK16E7 is one of the first oral immunotherapeutic vaccines that has been shown to have antineoplastic effects. Research on oral vaccines as a form of cancer treatment is preliminary but offers a new direction in terms of further treatment options. The use of bacteria in an antineoplastic treatment also offers encouraging new insight into future cancer treatment research. In an era of such high public vaccine skepticism, the need for alternative routes of administration may become paramount.

## 10. mRNA Vaccines

Messenger RNA (mRNA) vaccines have garnered significant attention since their role in combating the COVID-19 pandemic. Prior to the pandemic, mRNA vaccines were not widely used. This was in part due to the instability of unprotected mRNA when injected, as it would be quickly identified and destroyed by the host immune system. The solution to mRNA frailty has been found through the use of lipid nanoparticles, which coat and protect the vaccine and enhance delivery to antigen-presenting cells [48] Figure 2. The advantages of mRNA vaccines include high efficacy, the potential for rapid development, and a favorable safety profile [49]. Research is preliminary on how mRNA vaccines may be used as a therapeutic treatment for cancer; however, several ongoing studies show promise [50]. Considering the promise that mRNA has already shown, it should be further investigated as a therapeutic approach to treating cervical cancer.

## 11. Areas in Need of Continued Advancement

Cervical cancer vaccine research is a field filled with promising success and continued advancement [Table 2]. As new studies are completed, more areas in need of greater research become apparent.

Accum™ has potential as a vaccine delivery tool; however, more research needs to be performed on other vaccine formulations and other HPV strains to optimize its potential. Its success in treating HPV-16 should be applied to other strains of HPV, including HPV-18. This will provide new insights to see if Accum’s™ success can be expanded to treat multiple strains of HPV at once.

There have been multiple studies on DNA-based vaccines against cervical cancer looking at knocking out specific amino acid sequences. Future studies should look to find the most effective amino acids to mutate in order to maximize the effectiveness of cancer treatment.

Extensive work has already been carried out on generating successful treatment methods for HPV infections and cervical precancers. These treatments, as a form of combination therapy, should be investigated to determine whether they may be effective in treating advanced cervical cancer as well.

Research into oral and bacterial-based vaccines is preliminary, but the research has shown that they hold promise as cervical cancer treatments. More research is necessary to explore this potential. 

Immune checkpoint inhibitors have been shown to increase the efficacy of cervical cancer vaccines in multiple studies. Pembrolizumab targets PD-L1 and is the only FDA-approved immune checkpoint inhibitor for cervical cancer treatment. Since not all cervical cancers use PD-L1 as an immune checkpoint, further research is necessary to develop checkpoint inhibitors for other checkpoints. Tools such as transcriptome analysis may prove valuable in determining such interactions within the tumor microenvironment, and by identifying which immune checkpoints a patient’s tumor has repurposed and tailoring combination therapy to bypass that specific checkpoint, cervical cancer treatment could see higher success rates [51].

In the wake of the COVID-19 pandemic, mRNA vaccine potential has become apparent. As research improves our understanding of mRNA vaccines, their role in combatting cervical cancer must be fully explored.

## 12. Conclusions

With the introduction of HPV vaccines and screening programs, the world has seen a drastic decline in cervical cancer rates. However, the vaccines have not been perfected, and much of the world has yet to implement effective national vaccination and screening programs. Until HPV has been eradicated, patients will continue to be diagnosed with cervical cancer. As such, there will be a continued need for improved treatments for cervical cancer.

Considering how HPV infections affect the tissues of the cervix, vaccines may be the ideal treatment technique. New studies have shown advances in the use of Accum™ biotechnology, adenoviruses, and bacteria as delivery vectors. Protein-based and DNA-based vaccines have found success, with oral vaccines even emerging as a potential treatment option. The future of cervical cancer vaccine treatment is promising, offering hope to many still suffering from cervical cancer. Yet, much work must still be carried out to truly optimize the treatment potential of cervical cancer vaccines.

## Figures and Tables

**Figure 1 vaccines-13-00201-f001:**
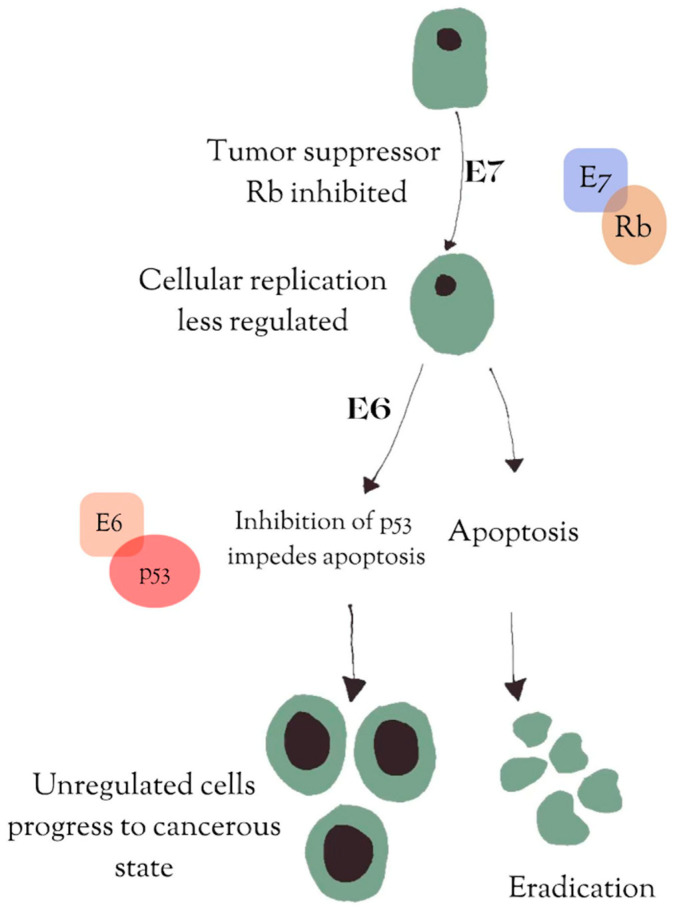
The early-region genes E6 and E7 play a large role in carcinogenesis. E6 inhibits the tumor suppressor gene p53, and E7 inhibits the tumor suppressor gene Rb. The inhibition of both p53 and Rb allows infected cells to avoid apoptosis and progress to a cancerous state.

**Figure 2 vaccines-13-00201-f002:**
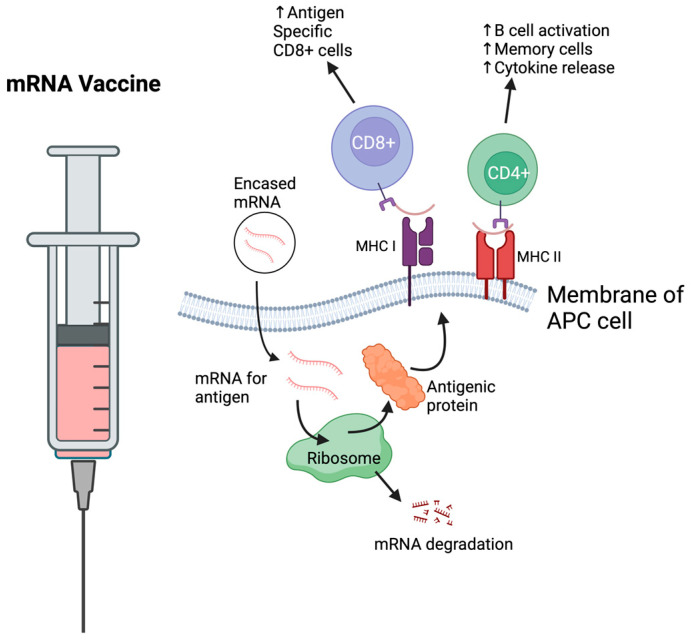
mRNA vaccines against cervical cancer would recognize an antigen within the cervical cancer and develop mRNA to produce an inactive antigenic protein. This would be encapsulated by a delivery system, which could increase mRNA delivery to APCs. The mRNA would travel to the ribosome, where it would be translated into its protein form. The MHC processing system would then cleave the antigenic protein and present it on its cell surface on both MHC class I and II molecules. This would be recognized by effector cells, which would initiate the immune response.

**Table 1 vaccines-13-00201-t001:** The FDA has approved three different prophylactic HPV vaccines over the years, with Gardasil 9 being the one currently approved for use in the United States. Though all vaccines offer high efficacy, none can guarantee complete immunity, justifying a need for continued research in cervical cancer treatment.

Vaccine Name (FDA Approved)	Target HPV Strains	Efficacy	FIDA Approval (Current vs. Previous)
Gardasil	6, 11, 16, 18	93-100%	Previous
Gardasil 9	6, 11, 16, 18, 31, 33, 45, 52, 58	97-100%	Current
Cervarix	16, 18	93%	Previous

**Table 2 vaccines-13-00201-t002:** Current therapeutic cervical cancer vaccine research is exploring a variety of strategies. Each vaccine strategy comes with its own unique advantages and should be fully investigated.

Vaccine Strategies	Unique Advantages
Accum^TM^ Vaccines	Increased vaccine delivery to APCs
Adenovirus Vector Vaccines	Historical success
Oral Vaccines	Ease of administration
mRNA Vaccines	High efficacy and rapid development

## Data Availability

The original contributions presented in this study are included in the article. Further inquiries can be directed to the corresponding author.

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
