# Peer review of "The Past, Present, and Future of Cervical Cancer Vaccines"

_vaccines, 2025, doi:10.3390/vaccines13020201_

Round 1
Reviewer 1 Report
Comments and Suggestions for Authors
dear authors,
I read with great interest the manuscript, which falls within the aim of this Journal. In my honest opinion, the topic is interesting enough to attract the readers’ attention. Nevertheless, authors should clarify some points and improve the discussion, as suggested below. Authors should consider the following recommendations:
In my opinion you have to improve the paper refering in the text to the updated literature on this topic comparing to otjer malignanicers as endometrial and breast cancer cancer,how its really important in these to preserv fertility before surgical or gonadotoxic treatment and refering yo other vaccines as covid ( zika previus pandemia) as well.
I suggest to read and cite these articles that surely improve the paper:
Fertility-Sparing Strategies for Early-Stage Endometrial Cancer: Stepping towards Precision Medicine Based on the Molecular Fingerprint
Symptomatic COVID-19 in Pregnancy: Hospital Cohort Data between May 2020 and April 2021, Risk Factors and Medicolegal Implications
CircuSymptomatic COVID-19 in Pregnancy: Hospital Cohort Data between May 2020 and April 2021, Risk Factors and Medicolegal Implications
Congenital Zika Syndrome: Genetic Avenues for Diagnosis and Therapy, Possible Management and Long-Term Outcomes
Human papillomavirus vaccine effectiveness by age at vaccination: A systematic review
Circulating miRNAs as a Tool for Early Diagnosis of Endometrial Cancer-Implications for the Fertility-Sparing Process: Clinical, Biological, and Legal Aspects
Sentinel Lymph Node Staging in Early-Stage Cervical Cancer: A Comprehensive Review
Measuring the composition of the tumor microenvironment with transcriptome analysis: past, present and future
Author Response
“In my opinion you have to improve the paper, referring in the text to the updated literature on this topic, comparing to other malignancies as endometrial and breast cancer, how it's really important in these to preserve fertility before surgical or gonadotoxic treatment and referring to other vaccines as covid (zika, previous pandemic) as well.”
Thank you for your insightful feedback and thoughtful suggestion. In response to your recommendation, we have reviewed our current sources, to ensure that they are both current and reliable sources. We agree that it would be interesting to compare cervical cancer malignancies to other malignancies such as endometrial and breast cancer. We would also like to thank you for sharing quality articles with us. We have reviewed them and have included several to strengthen our manuscripts integrity. This has undoubtedly improved the quality of our work.
Reviewer 2 Report
Comments and Suggestions for Authors
This is an interesting report on the status of HPV vaccines and uses as therapeutics. As a virologist and immunologist focusing on Herpesviruses and HIV, I have been confused and concerned by the medical approaches to Papilloma virus infections, specifically treatments after atypical cell pap smears and follow-up (or lack thereof) as well as the age restrictions for vaccination. Investigation of the literature suggests a lack of clinical studies on which to base treatments. I strongly agree with the authors that additional studies need to be conducted and look forward to progress in use of vaccines more extensively for prevention and treatment.
Having been asked by friends and family with concerns about HPV vaccines, my reading increased concerns about vaccine formulations, specifically the use of adjuvants and increases of dosage without regard to side effects. Adjuvants primarily increase antibody levels which is not the priority for a cell associated virus such as HPV and any immunologist who has extracted lymphoid organs from adjuvant treated mice is aware of the inflammation and adhesions adjuvants cause. Unfortunately, functional T-cell assays are difficult to perform (interferon gamma expression does not equal cytotoxicity), so the field tends to rely on antibodies as a read-out. However, it is good to see that T-cell interferon gamma expression was being used in the discussed studies.
Author Response
“This is an interesting report on the status of HPV vaccines and uses as therapeutics. As a virologist and immunologist focusing on Herpesviruses and HIV, I have been confused and concerned by the medical approaches to Papilloma virus infections, specifically treatments after atypical cell pap smears and follow-up (or lack thereof) as well as the age restrictions for vaccination. Investigation of the literature suggests a lack of clinical studies on which to base treatments. I strongly agree with the authors that additional studies need to be conducted and look forward to progress in use of vaccines more extensively for prevention and treatment.
Having been asked by friends and family with concerns about HPV vaccines, my reading increased concerns about vaccine formulations, specifically the use of adjuvants and increases of dosage without regard to side effects. Adjuvants primarily increase antibody levels which is not the priority for a cell associated virus such as HPV and any immunologist who has extracted lymphoid organs from adjuvant treated mice is aware of the inflammation and adhesions adjuvants cause. Unfortunately, functional T-cell assays are difficult to perform (interferon gamma expression does not equal cytotoxicity), so the field tends to rely on antibodies as a read-out. However, it is good to see that T-cell interferon gamma expression was being used in the discussed studies.”
Thank you for your thoughtful and constructive feedback. We are encouraged to hear that you also recognize the need for additional studies in vaccine research and development. Your perspective on adjuvant therapy as an immunologist is also invaluable. It is reassuring to learn that you approve of the research methods that were cited. Thank you for taking the time to help us ensure the quality and integrity of our work. It is much appreciated.
Reviewer 3 Report
Comments and Suggestions for Authors
The manuscript by Lien et al. is an excellent review of vaccines against HPV-causing cervical cancer. The information is very well organized and summarized, and the references are the most relevant to this research topic. It is a good contribution to the field and I believe it should be published in Vaccines. However, I have only one criticism that the authors shoud address.
RNA vaccines are safer than DNA vaccines. However, the authors do not mention whether there are RNA vaccines or whether any of those are in a trial phase. Anyway, I would like to know if would be possible to develop RNA vaccines and whether they would have any advantages over DNA-based vaccines.
Author Response
“RNA vaccines are safer than DNA vaccines. However, the authors do not mention whether there are RNA vaccines or whether any of those are in a trial phase. Anyway, I would like to know if it would be possible to develop RNA vaccines and whether they would have any advantages over DNA-based vaccines.”
Thank you for your insightful feedback. We greatly appreciate your suggestions, which have contributed to enhancing the clarity and depth of our review. In response to your recommendation, we have reviewed the literature and incorporated a section on the potential development of mRNA vaccines. Thank you again for your valuable input as it has undoubtedly improved the quality of our work.
Reviewer 4 Report
Comments and Suggestions for Authors
The current manuscript (ID: vaccines-3448305) titled "The Past, Present, and Future of Cervical Cancer Vaccines" provides a detailed overview of cervical cancer vaccines, including prophylactic and therapeutic options. However, there is a need for an extensive improvement in the manuscript to make it publishable in Vaccines.
1. The manuscript lacks the flowcharts and diagrams. Schematic diagrams illustrating vaccine mechanisms, immune responses, and HPV pathogenesis should be included.
2. Introduction and HPV sections restate similar information about HPV transmission and oncogenesis. The authors need to condense redundant information and focus more on vaccine innovations.
3. The manuscript highlights successes but does not fully address challenges, such as vaccine accessibility, global disparities, and immune escape mechanisms. Include a section on real-world challenges, such as cost, availability in low-income countries, and vaccine hesitancy.
4. The authors need to add a table comparing prophylactic and therapeutic vaccines in terms of effectiveness, immune response, and clinical trial outcomes.
5. The authors should incorporate more recent literature, particularly from high-impact journals on mRNA-based vaccines and combination therapies.
Author Response
“1. The manuscript lacks the flowcharts and diagrams. Schematic diagrams illustrating vaccine mechanisms, immune responses, and HPV pathogenesis should be included.”
Thank you for providing feedback on the manuscript's figures. The current manuscript contains a figure (Figure. 1) that provides a visual interpretation of HPV oncogenesis. Additionally, we have added another figure (Figure. 2) considering the concerns you raised. Thank you again for verifying the quality of our work as it has greatly improved the quality of our paper.
“2. Introduction and HPV sections restate similar information about HPV transmission and oncogenesis. The authors need to condense redundant information and focus more on vaccine innovations.”
Thank you for your acute attention to detail. We agree that the focus of the paper should be more on vaccine innovation and less on redundant HPV background information. In response to your suggestion, we have analyzed the manuscript to cut down on redundant information and focus on more important material.
“3. The manuscript highlights successes but does not fully address challenges, such as vaccine accessibility, global disparities, and immune escape mechanisms. Include a section on real-world challenges, such as cost, availability in low-income countries, and vaccine hesitancy.”
Thank you for bringing this to our attention. We are grateful for your efforts. We agree that the current manuscript does not address all the current challenges that vaccine development faces currently. As such, we have added some content on a few of the many hurdles that stand in the way of ideal vaccine production. Again, we are very thankful for your feedback, as it helps us improve the manuscript.
“4. The authors need to add a table comparing prophylactic and therapeutic vaccines in terms of effectiveness, immune response, and clinical trial outcomes.”
Thank you for this insightful feedback. We agree that a table would be of great value in our manuscript, and as such, have included Table. 1 to illustrate prophylactic HPV vaccines. To address your concerns, we have also developed Table. 2, to make a simple comparison between vaccine strategies. Though we would have preferred a table that could compared vaccine efficacy, we are unable to produce one at this time. We hope that the current addition is sufficient. Your time and efforts are extremely valuable to us, and we thank you again for such insightful feedback.
“5. The authors should incorporate more recent literature, particularly from high-impact journals on mRNA-based vaccines and combination therapies.”
Thank you for the constructive feedback. We agree that the manuscript should incorporate the most recent literature from high-impact journals. In response to your recommendation, we have reviewed the literature and included a section on mRNA. This addition has undoubtedly improved the quaity of our paper and we are incredibly grateful to you for your assistance in the process.
Round 2
Reviewer 1 Report
Comments and Suggestions for Authors
now the paper is suitable for pubblication
Reviewer 4 Report
Comments and Suggestions for Authors
The authors have responded to my comments in the revised manuscript.